# DFA-VLA: Enhancing Robotic Manipulation via Embodied Intelligence

## Abstract

With the rapid advancement of robotic hardware and software technologies, embodied intelligence has become pivotal, enabling physical agents to interact with the environment in real-time via multimodal inputs and make autonomous decisions through a closed-loop sensor-actuator system. Among mainstream methods, end-to-end Vision-Language-Action (VLA) models efficiently execute robotic tasks by directly mapping perception to actions but suffer from critical limitations: poor modeling of fine-grained visual elements (e.g., occluded regions, small objects) and over-reliance on static cross-modal attention, restricting adaptability and generalization in complex open environments. To address these, this paper focuses on enhancing task execution accuracy, timeliness, and generalization via embodied intelligence, with a core innovation in the Dynamic Fine-grained Alignment-based Vision-Language-Action (DFA-VLA) model built on a pre-trained large language model backbone. It integrates two key modules: the Multi-scale Visual-Semantic Modeling (MVSM) Module, which combines a vision transformer and a segment anything model to extract high-resolution semantic features, using semantic masks to boost perception of small objects, occlusions, and cluttered backgrounds (with replaceable encoders for scene adaptation); and the Dynamic Fine-grained Alignment and Fusion (DFAF) Module, which employs mask-guided sparse dynamic attention for efficient language-visual alignment (reducing redundant computations) and a dynamic gating network (via text semantics) to adaptively switch between vision- and language-driven strategies. Both evaluations on LIBERO benchmarks and real-world settings show that DFA-VLA outperforms state-of-the-art methods, especially in spatial reasoning and long-term tasks, with higher success rates and inference efficiency. Parameter-efficient fine-tuning (e.g., LoRA) reduces resource use for task/hardware adaptation, while a Sim2Real pipeline validates real-world effectiveness on physical robots, confirming improved generalization in unstructured scenarios.

## 1 Introduction

In recent years, technological innovations in the field of artificial intelligence have laid a solid foundation for the vigorous development of embodied intelligence. From the rise of large language models (LLMs) based on the Transformer architecture, such as GPT-4 OpenAI (2023), BERT Devlin et al. (2018), and T5 Raffel et al. (2019), to the rapid evolution of vision-language models (VLMs), a series of breakthroughs have created the necessary conditions for the implementation of embodied intelligence.

Large Language Models (LLMs) support natural language processing (NLP) with their powerful semantic understanding and generation capabilities, while Vision-Language Models (VLMs) break down modal barriers through techniques such as cross-attention mechanisms and contrastive learning, promoting the development of cross-modal cognition. Against this backdrop, end-to-end vision-language-action models like OpenVLA John D, Jane S, Alice J, et al. (2023) , RT-2 Anthony B, Noah B, Justice C, et al. (2023) and PaLM-E Anthony B, Michael A, Joseph M, et al. (2022) seek to enable robots to integrate environmental perception (via RGB or video), language understanding, and action execution into a unified framework. These unified VLA frameworks based on pre-trained VLMs or LLMs embodies the fundamental workflow of embodied intelligence models for fixed robots, from high-level planning to low-level control, and establishes a core paradigm for the tech-

nical implementation of embodied intelligence. However, these models are limited by insufficient fine-grained visual modeling and static cross-modal interaction mechanisms, restricting their adaptability and generalization in complex unstructured environments. To address these limitations, this paper proposes a vision-language-action integrated model framework based on dynamic fine-grained alignment, aiming to enhance task execution accuracy, timeliness, and generalization ability of embodied intelligent systems in complex real-world scenarios.

## 2 RELATED WORK

**Classical Manipulation and Vision-Language-Action Models.** Traditional robotic manipulation approaches relied on sampling-based motion planning (e.g., RRT/RRT* LaValle (1998); Karaman & Frazzoli (2011)) for collision-free trajectory generation and rule-based systems for repetitive tasks. While effective in structured environments, these methods require precise goal specifications and cannot interpret semantic instructions, motivating the development of end-to-end vision-language-action (VLA) frameworks. Recent VLAs such as RT-2, Octo, and OpenVLA Anthony B, Noah B, Justice C, et al. (2023); Michael B, Emily D, Robert W (2023); John D, Jane S, Alice J, et al. (2023) align visual tokens with language representations to generate robotic actions, demonstrating real-world viability. However, their static cross-modal attention mechanisms tend to over-attend to irrelevant visual regions, struggle with dynamic modality balance adjustment, and exhibit limited generalization in cluttered scenes and long-horizon tasks. We address these limitations through mask-guided sparse attention and learnable dynamic gating that adaptively balances vision-language fusion based on task context.

**Fine-Grained Visual Modeling.** Multi-scale architectures (FPN, PANet Lin et al. (2017); Liu et al. (2018)) and vision transformers (Swin Liu et al. (2021)) have advanced feature extraction, while segmentation models like SAM Kirillov et al. (2023) provide precise object masks for localizing small or occluded objects. Yet prior work rarely couples such masks with dynamic cross-modal routing or task-adaptive computation allocation. Our MVSM module exploits DinoV2 Oquab et al. (2023) for global scene context alongside SAM-guided local regions, while our DFAF module concentrates sparse attention on task-critical areas—addressing the efficiency-accuracy trade-off in fine-grained perception.

**Sim2Real Transfer and Model Adaptation.** Sim2Real transfer faces persistent challenges from visual domain gaps (rendering artifacts, sensor noise), dynamics mismatches (idealized friction, contact modeling), and multi-modal misalignments that impede generalization to long-horizon, semantically complex tasks Höfer et al. (2021); Ho et al. (2021); Leung et al. (2022). Concurrently, parameter-efficient fine-tuning (PEFT) methods like LoRA Team (2023); Liu et al. (2023b) enable task adaptation with minimal trainable parameters but suffer performance degradation under quantization, with modality-specific sensitivities (e.g., vision encoders being more sensitive than language decoders) complicating deployment Frantar et al. (2022); Kulkarni et al. (2024). We validate our approach in both LIBERO simulation and physical robots, employing PEFT with mixed-precision inference to balance efficiency and accuracy across diverse manipulation scenarios.

## 3 METHODOLOGY

This section presents our methodology for addressing key challenges in robotic manipulation. We focus on three core components: (1) a novel vision-language-action (VLA) model architecture with fine-grained perception capabilities, (2) parameter-efficient adaptation mechanisms for cross-task generalization, and (3) a Sim2Real transfer pipeline designed to mitigate domain discrepancies.

**Model Architecture.** We propose a Dynamic Fusion Adapter VLA (DFA-VLA) model, which integrates mask-guided sparse attention and dynamic modality gating to prioritize task-critical features. Specifically, the model takes $224 \times 224$ RGB images and natural language instructions (e.g., manipulation commands) as inputs, and outputs pose adjustment vectors $[\Delta x, \Delta \theta, \Delta \text{Grip}]$ for robotic end-effectors (encoding translational movement, rotational orientation, and gripper actuation). It comprises four core components: a Multimodal Visual Sensing Module (MVSM) for hierarchical scene perception (combining DinoV2 global context and SAM-driven local object features), a layered text encoder (leveraging Llama 2 to extract syntactic-semantic hierarchies from instructions), a Dynamic Fusion and Attention Filtering (DFAF) module for adaptive cross-modal feature align-

ment (via sparse similarity sorting and gating mechanisms), and an action de-tokenizer that maps fused representations to executable robotic motions. The complete workflow of these interconnected modules is visualized in Figure 1 and Figure 2.

The Multimodal Visual Sensing Module (MVSM) extracts hierarchical visual features for cross-modal fusion, formalized as follows: Given an input RGB image $\mathbf{I} \in \mathbb{R}^{H \times W \times 3}$ (with $H = W = 224$), a pre-trained DinoV2 modelOquab et al. (2023) generates a global scene feature map $\mathbf{V}_{\text{global}} \in \mathbb{R}^{S \times S \times d_v}$ (where $S = 14$, $d_v$ is the visual feature dimension), capturing both broad context (e.g., background layout) and fine-grained patterns. This 2D map is flattened into a 1D sequence via

$$\text{Flatten}(\mathbf{V}_{\text{global}}) \in \mathbb{R}^{S^2 \times d_v} \tag{1}$$

since $S^2 = 14^2 = 196$. Simultaneously, the Segment Anything Model (SAM)Kirillov et al. (2023) performs instance segmentation, outputting a set of bounding boxes $\mathcal{B} = \{\text{box}_1, \text{box}_2, \dots, \text{box}_k\}$; Region of Interest Alignment (RoIAlign) then crops task-relevant regions from $\mathbf{V}_{\text{global}}$, yielding local object features

$$\mathbf{V}_{\text{local}} = \text{RoIAlign}(\mathbf{V}_{\text{global}}, \mathcal{B}) \in \mathbb{R}^{k \times d_v} \tag{2}$$

emphasizing fine-grained attributes like object contours, with $k$ as the number of task-related objects. These global and local features are vertically concatenated:

$$\mathbf{V}_{\text{concat}} = [\text{Flatten}(\mathbf{V}_{\text{global}}); \mathbf{V}_{\text{local}}] \in \mathbb{R}^{(S^2+k) \times d_v} \tag{3}$$

where $[\cdot; \cdot]$ denotes vertical concatenation. To unify visual and language feature spaces, a linear projection maps $\mathbf{V}_{\text{concat}}$ to the text encoder's dimension $d_t$ (e.g., $d_t = 4096$ for Llama 2):

$$\mathbf{V}_{\text{proj}} = \mathbf{W}_{\text{proj}} \cdot \mathbf{V}_{\text{concat}} + \mathbf{b}_{\text{proj}} \in \mathbb{R}^{(S^2+k) \times d_t} \tag{4}$$

with projection matrix $\mathbf{W}_{\text{proj}} \in \mathbb{R}^{d_t \times d_v}$ and bias $\mathbf{b}_{\text{proj}} \in \mathbb{R}^{d_t}$. This design addresses traditional limitations of overlooked scene context or ignored object details, enabling robotic manipulation tasks that require both semantic reasoning (via language) and physical precision (via visual cues).

On the other hand, the raw natural language instruction $\mathbf{I}_{\text{lang}}$ is first tokenized by the Llama tokenizer into a sequence of $L$ tokens $\{t_i\}_{i=1}^{L}$, which are then embedded via

$$\mathbf{T} = [\mathbf{e}_{t_1}; \mathbf{e}_{t_2}; \dots; \mathbf{e}_{t_L}] \in \mathbb{R}^{L \times d_t} \tag{5}$$

$$\mathbf{e}_{t_i} = \text{Embedding}(t_i) \in \mathbb{R}^{d_t} \tag{6}$$

where $d_t$ is the text embedding dimension.

Next, the projected visual features $\mathbf{V}_{\text{proj}} \in \mathbb{R}^{(S^2+k) \times d_t}$ are vertically concatenated with the text embeddings to form the initial multimodal input (Figure 1):

$$\mathbf{X} = \begin{bmatrix} \mathbf{T} \\ \mathbf{V}_{\text{proj}} \end{bmatrix} \in \mathbb{R}^{(S^2+k+L) \times d_t} \tag{7}$$

This combined representation $\mathbf{X}$ serves as input to our DFAF module for selective cross-modal attention and fusion (Figure 2). The MVSM module enables robust multi-scale feature extraction and pixel-level segmentation, accurately capturing both detailed local features and global scene context to improve perception in diverse scenarios.

Given $\mathbf{X}$, we feed it into the 32-layer Transformer decoder of Llama2 7B **?**. For the first three blocks, we apply our DFAF module to refine $H_{\text{text}} = \mathbf{H}_{1:L}^3 \in \mathbb{R}^{L \times d_t}$ and sparsify the visual tokens down to the top-$k$ regions before reinjecting. Concretely, DFAF computes bidirectional attention and a gating fusion:

$$\alpha = \sigma\big(W \, \text{Mean}(H_{\text{text}})\big) \tag{8}$$

$$V_{\text{sparse}} = \text{TopK}\big(\cos(H_{\text{text}}, \mathbf{V}_{\text{proj}})\big) \tag{9}$$

$$F = \alpha \odot V_{\text{sparse}} + (1 - \alpha) \odot \text{Broadcast}\big(\text{Mean}(H_{\text{text}})\big) \tag{10}$$

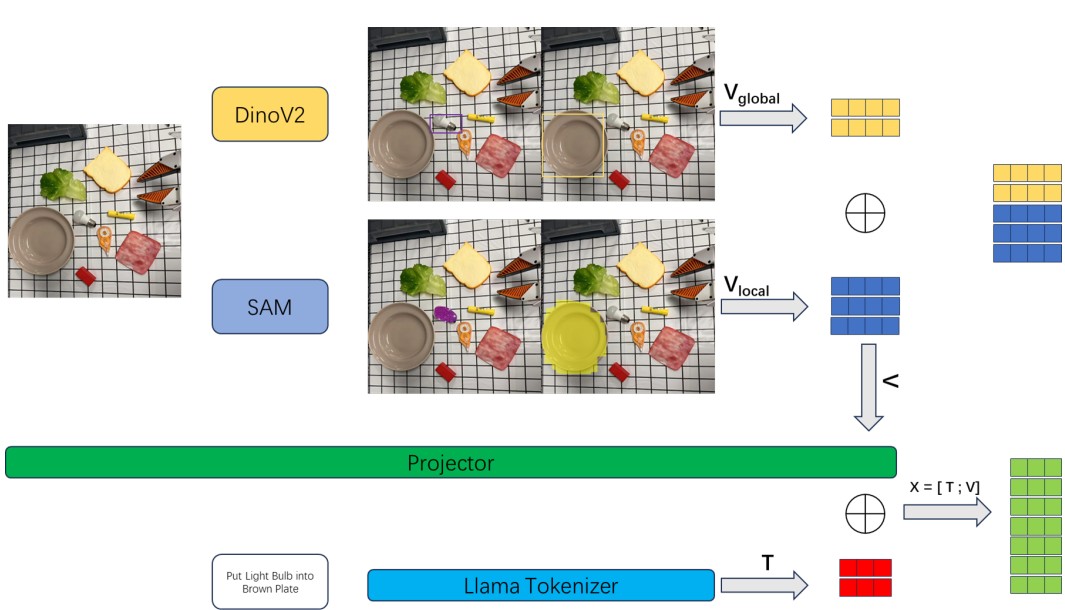

Figure 1: The Structure of Multimodal Visual Sensing Module (MVSM). Multimodal feature modeling and extraction.

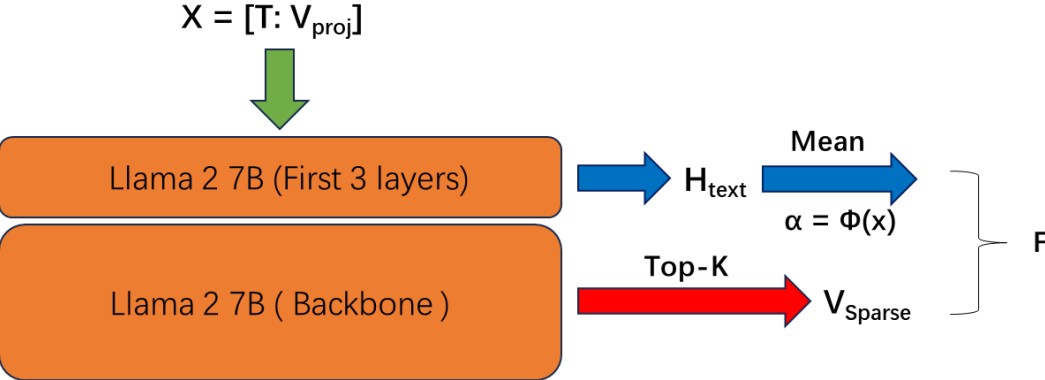

Figure 2: The Structure of the Dynamic Fine-grained Alignment and Fusion Module (DFAF). Sparse attention mechanism and feature fusion gating.

From layer 4 to 32, the model performs deep semantic reasoning solely on the fused sequence without further visual pruning. This design (Figure 2) ensures that fine-grained region-level alignments are learned in shallow layers while deep layers focus on high-level task planning.

**Action De-Tokenizer.** The final feature $F$ is then projected onto the robot action vocabulary and decoded into a sequence of executable commands. Mathematically, the action at time step $t$ ($a_t$) is generated by policy $\pi_\theta$ acting on feature $F$:

$$a_t = \pi_\theta(F) \tag{11}$$

Concretely, $a_t$ comprises translational displacement $\Delta x_t$, angular displacement $\Delta \theta_t$, and gripping action $\text{Grip}_t$:

$$a_t = (\Delta x_t, \Delta \theta_t, \text{Grip}_t) \tag{12}$$

## 4 EXPERIMENTS

### 4.1 DATASET AND SPLITTING STRATEGY

To ensure reproducibility and rigorous evaluation, we describe the datasets, preprocessing, and splitting protocol used in our experiments.

**LIBERO Simulation Dataset.** We use the official LIBERO benchmark Liu et al. (2023a), a standard suite for tabletop robotic manipulation. LIBERO is partitioned into four subsets: *LIBERO-Spatial*, *LIBERO-Object*, *LIBERO-Goal*, and *LIBERO-Long*, which together cover short-horizon spatial reasoning, novel-object manipulation, goal-conditioned tasks, and long-horizon multi-step tasks. Models are trained on the provided training demonstrations and evaluated on the held-out test episodes from the benchmark to ensure strict separation between training and evaluation.

Visual observations are resized to $224 \times 224$, normalized with ImageNet statistics, and represented as a 3-frame RGB stack (depth, when used, is normalized and concatenated). Actions are parameterized as 7-D operational commands (3D position, 3D orientation, binary gripper), scaled to actuator ranges used in the simulator.

**BridgeData V2 Pre-training.** We initialize the visual backbone using pretraining on BridgeData V2 Mu et al. (2023) and DinoV2 Oquab et al. (2023). BridgeData V2 is used solely for representation learning and *not* for downstream LIBERO fine-tuning or test evaluation; exact pretraining protocols and dataset statistics are provided in the appendix.

**Real-world Validation.** Physical validation is performed by deploying fine-tuned policies directly onto industrial robotic platforms (Zhichang CR5 and Lebai LB-10) without collecting additional real-world training data, i.e., we evaluate zero-shot sim-to-real transfer. Physical tasks are chosen to be semantically analogous to LIBERO families and are adapted to the workspace and available objects. Quantitative and qualitative evaluation protocols for real-world tests are detailed in the appendix.

**Preprocessing and Splits.** We apply standard image augmentations (random crop/resize, small rotation, brightness/contrast jitter) and preserve canonical train/validation/test splits where available. All custom splits (for ablation or hyperparameter studies) are constructed to avoid overlap in episodes or object instances with the test set. Random seeds and exact split manifests are released in the supplementary materials to facilitate reproducibility.

### 4.2 EXPERIMENTAL PLATFORM DETAILS

We run experiments in both simulation and on physical platforms, using consistent software and hardware stacks to ensure fair comparisons. Implementation specifics (data augmentation parameters, hyperparameter grids, model architecture details, and training acceleration settings such as LoRA/INT8/AMP/FlashAttention/FSDP) are provided in Appendix **??**.

The detailed **Experimental Setup** (task selection, trial counts, baselines, and evaluation metrics) is described next.

**Experimental Setup.** Following established protocols John D, Jane S, Alice J, et al. (2023), we conduct experiments on both the LIBERO simulation platform Liu et al. (2023a) and physical robotic

platforms featuring Zhichang CR5 and Lebai LB-10 robotic arms Emergen (2024); LEBAI (2022). For LIBERO simulation, we evaluate on four task subsets: LIBERO-Spatial, LIBERO-Object, LIBERO-Goal, and LIBERO-Long, following standard evaluation protocols in robotic manipulation research. We predefine 50 representative operation tasks across these subsets and implement our approach (DFA-VLA fine-tuned) for 10 iterations to ensure result reliability, with each iteration running at a 1-second time granularity. For baseline comparisons, we include Diffusion Policy Pathak et al. (2023) from scratch, Octo Michael B, Emily D, Robert W (2023) fine-tuned, and OpenVLA John D, Jane S, Alice J, et al. (2023) fine-tuned models, with all policies evaluated across Success Rate (SR) and Rank metrics, alongside inference Latency (ms) for efficiency analysis and Parameter Count for model complexity assessment. On physical platforms, tests are executed in real time with equivalent task configurations to validate simulation transferability.

To evaluate hyperparameter sensitivity, we perform dedicated experiments on the LIBERO-Spatial subset, testing key parameters including $\lambda_1$ (0.1, 0.5, 1.0, 2.0), $\lambda_2$ (0.1, 0.5, 1.0), input resolution (128×128, 224×224, 512×512), Top-K values (2, 5, 10), and gating mechanisms (fixed $\alpha$=0.0, fixed $\alpha$=1.0, learnable $\alpha$). These hyperparameters are directly motivated by the design of the DFA-VLA model and its multi-objective optimization framework: the weighting terms $\lambda_1$ and $\lambda_2$ are used to balance the action prediction loss $\mathcal{L}$action, cross-modal alignment loss $\mathcal{L}$align, and semantic region selection loss $\mathcal{L}$mask (Equation 16). The Top-K value controls how many high-attention regions are selected for supervision in $\mathcal{L}$mask , while the gating mechanism (parameterized by $\alpha$) determines the aggregation strategy of multi-region features in the perception module.

$$\mathcal{L}_{\text{total}} = \mathcal{L}_{\text{action}} + \lambda_1 \mathcal{L}_{\text{align}} + \lambda_2 \mathcal{L}_{\text{mask}} \tag{13}$$

$$\mathcal{L}_{\text{action}} = \begin{cases} 0.5(\hat{a}_t - a_t)^2, & \text{if } |\hat{a}_t - a_t| < 1 \\ |\hat{a}_t - a_t| - 0.5, & \text{otherwise} \end{cases} \tag{14}$$

$$\mathcal{L}_{\text{align}} = -\log \frac{\exp(\text{sim}(t, v^+)/\tau)}{\sum_{i=1}^{k} \exp(\text{sim}(t, v_i)/\tau)} \tag{15}$$

$$\mathcal{L}_{\text{mask}} = -\frac{1}{K} \sum_{i=1}^{K} \log \alpha_i \tag{16}$$

Also, to further analyze the contribution of each model module in DFA-VLA, we design ablation experiments based on the LIBERO simulation environmentLiu et al. (2023a).As mentioned earlier, DFA-VLA includes two innovative modules: MVSM (Multi-scale Visual-Semantic Modeling) and DFAF (Dynamic Feature Aggregation). These modules are designed to improve the model's performance in polymorphic tasks, especially in complex spatial relationships and long-term task scenarios. By systematically removing or adjusting these modules, we evaluate their specific impact on Success Rate (SR), Rank, Latency, and Parameter Count.

We still use the LIBERO-Spatial task suite as the main test scenario and design five groups of experiments, gradually introducing or removing the MVSM and DFAF modules to observe their impact on model performance. Each group of experiments is evaluated in 500 trials, based on three random seeds to ensure the stability of the results. The experimental results are summarized in Table 3. The five model variants are defined as follows:

- Model 1: Basic structure configuration (Llama tokenizer + ViT + no gating)
- Model 2: Add MVSM module to Model 1
- Model 3: Add DFAF module to Model 1
- Model 4: Add both MVSM and DFAF modules to Model 1 (without gating)
- Model 5: Add both MVSM and DFAF modules to Model 1 (Full DFAF with gating)

The specific structure of the basic model(Model 1)is as follows: Input layer processes text via LLaMA tokenizer and 224×224 RGB images via ViT-BaseDosovitskiy et al. (2020). Feature extraction uses LLaMA-2 7B Touvron et al. (2023) for text and a 3-layer CNNLong et al. (2015) for

visuals. Fusion projects features to a shared space and combines them element-wise. Output layer uses a 2-layer Transformer decoder to generate 7D action commands (position, orientation, gripper control) with tanh activation for operational bounds.

For the physical experiments, we utilize two types of robotic arms: the Zhichang CR5 and the Lebai LB-10 Emergen (2024); LEBAI (2022), both equipped with 6 degrees of freedom and end-effectors (e.g., grippers). These robots enable precise motion control for executing manipulation tasks. The experimental environment is configured as a standardized workspace (1.2 m × 0.8 m × 0.7 m), including a workbench, adjustable lighting (500–800 lux), and overhead RGB-D cameras (Intel RealSense D435i and GoPro) Intel (2024); GoPro (2024). Various objects—such as plastic fruits, toy dinosaurs, and boxes—are provided to facilitate task diversity and test generalization of robotic behaviors.

In this study, based on the BridgeData V2 datasetMu et al. (2023), we realize the efficient deployment of the model in the inference stage through LoRAHu et al. (2021) fine-tuning and INT8Banner et al. (2018) quantitative inference technology. In the training process, the combination of automatic mixed accuracy (AMP)Micikevicius et al. (2017), FlashAttentionDao et al. (2022) and FSDP multi-GPU parallelZhao et al. (2021) strategies significantly improves the efficiency and stability of large model training. Based on 22 standard tasks in BridgeData V2, we cover five types of generalization capabilities for visual, motor, physical, semantic, and language instructions, and repeat the experiment 10 times for each task, using a three-level scoring system of success (1 point), partial success (0.5 points), and failure (0 points).

# 5 RESULTS AND DISCUSSION

We evaluate DFA-VLA through comprehensive experiments across simulation benchmarks and real-world robotic platforms. Our results demonstrate consistent improvements over state-of-the-art baselines across multiple task categories and evaluation metrics.

Table 1: Performance comparison of DFA-VLA on LIBERO Tasks. SR and Rank over 3 seeds.

| Policy | LIBERO-Spatial | | LIBERO-Object | | LIBERO-Goal | | LIBERO-Long | | Average | |
|---|---|---|---|---|---|---|---|---|---|---|
| | SR (↑) | Rank (↓) | SR (↑) | Rank (↓) | SR (↑) | Rank (↓) | SR (↑) | Rank (↓) | SR (↑) | Rank (↓) |
| Diffusion Policy | $78.3 \pm 1.1\%$ | 3 | $92.5 \pm 0.7\%$ | 3 | $68.3 \pm 1.2\%$ | 3 | $50.5 \pm 1.3\%$ | 3 | $72.4 \pm 0.7\%$ | 2.5 |
| Octo fine-tuned | $78.9 \pm 1.0\%$ | 2 | $85.7 \pm 0.9\%$ | 3 | $84.6 \pm 0.9\%$ | 1 | $51.1 \pm 0.6\%$ | 2 | $75.1 \pm 0.6\%$ | 2 |
| OpenVLA fine-tuned | $84.7 \pm 0.9\%$ | 1 | $88.4 \pm 0.8\%$ | 2 | $79.2 \pm 1.0\%$ | 2 | $53.7 \pm 0.6\%$ | 1 | $76.5 \pm 0.6\%$ | 1.5 |
| **DFA-VLA fine-tuned** | $\mathbf{85.5 \pm 0.9\%}$ | **1** | $\mathbf{89.8 \pm 0.8\%}$ | **1** | $80.2 \pm 1.0\%$ | 2 | $\mathbf{55.5 \pm 0.6\%}$ | **1** | $\mathbf{77.8 \pm 0.6\%}$ | **1.25** |

As for hyperparameter analysis, Table 2 reveals that $\lambda_1$ and $\lambda_2$ at 0.5 yield the optimal SR (85.5%) and Rank (1.2), with higher values causing performance drops due to over-regularization. Input resolution tests show 224×224 balances accuracy and latency (60 ms), as 512×512 only marginally improves SR (85.6%) but increases latency by 20%. Top-K=5 is ideal, with slight declines for smaller or larger values, while learnable $\alpha$ in gating outperforms fixed values by 2.1-2.6 percentage points in SR, validating adaptive weighting.

The next one is ablation Experiments. As shown in Table 3, demonstrate that the baseline Model 1 (82.3% SR, Rank 3.1) benefits significantly from module additions: MVSM alone boosts SR by 2.4 points (84.7%, Rank 1.9) by enhancing spatial understanding, while DFAF alone improves SR by 1.7 points (84.0%, Rank 2.3) via efficient feature aggregation. Combining both modules without gating (Model 4) achieves 85.1% SR (Rank 1.5), and adding adaptive gating (Model 5) further raises SR to 85.5% (Rank 1.2), confirming synergistic effects between MVSM, DFAF, and gating.

Table 2: Hyperparameter Sensitivity Analysis on LIBERO-Spatial. We report Success Rate (SR), Rank, and Latency across different parameter settings. Input resolution refers to RGB image dimensions.

| Hyperparameter | SR (↑) | Rank (↓) | Latency (ms, ↓) |
|---|---|---|---|
| $\lambda_1 = 0.1$ | 82.1% | 2.8 | 58 |
| $\lambda_1 = 0.5$ | **85.5%** | **1.2** | 60 |
| $\lambda_1 = 1.0$ | 85.0% | 1.3 | 62 |
| $\lambda_1 = 2.0$ | 83.9% | 2.1 | 65 |
| $\lambda_2 = 0.1$ | 84.8% | 1.4 | 60 |
| $\lambda_2 = 0.5$ | **85.5%** | **1.2** | 60 |
| $\lambda_2 = 1.0$ | 84.9% | 1.3 | 61 |
| $128 \times 128$ | 84.2% | 2.0 | 52 |
| $224 \times 224$ | 85.5% | 1.2 | 60 |
| $512 \times 512$ | **85.6%** | **1.1** | 72 |
| Top-$K = 2$ | 85.2% | 1.3 | 58 |
| Top-$K = 5$ | **85.5%** | **1.2** | 60 |
| Top-$K = 10$ | 85.3% | 1.3 | 62 |
| Gating: $\alpha$ fixed = 0.0 | 82.9% | 2.5 | 57 |
| Gating: $\alpha$ fixed = 1.0 | 83.4% | 2.2 | 58 |
| Gating: learnable $\alpha$ | **85.5%** | **1.2** | 60 |

Table 3: Ablation study results on LIBERO-Spatial. We report the individual contributions of MVSM and DFAF modules across Success Rate (SR), Rank, Latency.

| ID | MVSM | DFAF | SR (↑) | Rank (↓) | Latency (ms) | Params (M) |
|---|---|---|---|---|---|---|
| 1 | | | 82.3% | 3.1 | 55 | 78.5 |
| 2 | ✓ | | 84.7% | 1.9 | 62 | 146.3 |
| 3 | | ✓ | 84.0% | 2.3 | 58 | 83.2 |
| 4 | ✓ | ✓ (w/o gate) | 85.1% | 1.5 | 60 | 149.6 |
| 5 | ✓ | ✓ (Full DFAF) | **85.5%** | **1.2** | 60 | 151.2 |

Finally, we report the results of the robotic arm in real-world scenarios. Per-task results (22 tasks across 5 categories) are provided in Appendix A (Table 4); Table 1 summarizes benchmark SOTA in simulation.

## 6 CONCLUSION AND FUTURE WORK

This work addresses critical limitations of existing Vision-Language-Action (VLA) models in robotic manipulation, particularly their poor handling of fine-grained visual elements (e.g., occlusions, small objects) and rigid cross-modal alignment. We propose the Dynamic Fine-grained Alignment-based VLA (DFA-VLA) model, which integrates two key innovations: (1) the Multi-scale Visual-Semantic Modeling (MVSM) module, fusing high-resolution visual features with semantic masks to enhance perception of complex visual scenarios; and (2) the Dynamic Fine-grained Alignment and Fusion (DFAF) module, leveraging mask-guided sparse attention and text-semantic-driven dynamic gating to enable adaptive cross-modal alignment.

Future work will advance Vision-Language-Action (VLA) models in embodied robotics through four interconnected directions: first, expanding experimental scenarios to include real-world tasks with high semantic ambiguity and complex combinatorial structures (e.g., "stacking fragile objects on soft cushions" or "sorting by color-shape rules") to better showcase the model's ability

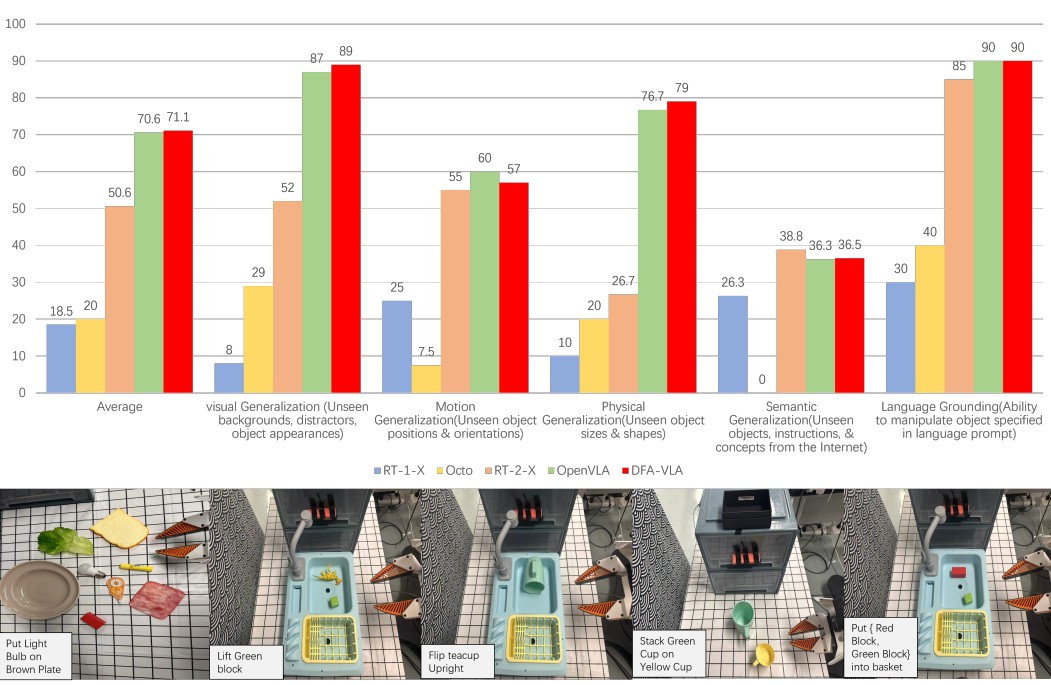

Figure 3: BridgeData V2 robot evaluation tasks and results. The task success rate (SR) of the DFA-VLA model under 22 subtasks in 5 scenario classifications. Average success rates ± StdErr are computed across 220 total rollouts per approach.

to link language understanding and action execution, addressing current limitations in object diversity and task complexity; second, moving beyond incremental optimizations of existing architectures (e.g., CLIPort, OpenVLA John D, Jane S, Alice J, et al. (2023)) to explore foundational innovations, such as building unified task abstraction structures (e.g., "task grammar trees") and reconfiguring the perception-understanding-action pipeline around task logic, while breaking static cross-modal fusion patterns with dynamic interaction mechanisms; third, enhancing real-world utility and human-robot collaboration by integrating large language models and active learning, enabling robots to autonomously clarify ambiguous instructions (e.g., "Do you mean placing A into B?") and develop self-supervised review abilities for multi-turn tasks (e.g., "I completed the first step; what's next?"); and finally, exploring cross-task, cross-scene, and cross-modal generalization through multi-domain pretraining (e.g., kitchen operations, indoor cleaning) with meta-learning and multimodal self-supervision, while unifying task levels via a universal language interface to generate complete task execution graphs from single sentences.

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

# A ADDITIONAL RESULTS

Table 4: Real-world manipulation performance across 22 tasks. We report success rates (out of 10 trials per task) across five generalization categories. Higher scores indicate better performance.

| Category | Task | # Trials | RT-1-X | Octo | RT-2-X | OpenVLA | DFA-VLA |
|---|---|---|---|---|---|---|---|
| **Visual Gen** | Put Light Bulb on Brown Plate (Easy) | 10 | 1 | 5 | 7 | 10 | **10** |
| | Put Light Bulb on Brown Plate (Cluttered) | 10 | 0 | 1 | 5 | 9 | **9.5** |
| | Put Teacup from Basket into Sink | 10 | 1 | 0 | 0 | 7 | **7** |
| | Put Apple into Bowl (w/ Clutter) | 10 | 1 | 3.5 | 6 | 6.5 | 6 |
| | Put Yellow Bread on Pink Plate | 10 | 1 | 4 | 8 | **9** | 8.5 |
| | Put Yellow Bread on Pink Plate (Shiny) | 10 | 1 | 3 | 7 | 7 | **8** |
| **Motion Gen** | Lift Green Block | 10 | 3 | 0.5 | 6.5 | **7.5** | 7 |
| | Put Apple on Brown Plate (Height Change) | 10 | 2 | 1 | 4.5 | 4.5 | 4.5 |
| | Stack Red Cube on Green Cube | 10 | 2 | 0 | 8 | 9 | **10** |
| | Move Spoon from Table to Mug | 10 | 4 | 1 | 7 | 7 | 7 |
| **Physical Gen** | Put Apple on Brown Plate | 10 | 1 | 0 | 1 | 7.5 | **8** |
| | Flip Teacup Upright | 10 | 2 | 6 | 4.5 | 8 | 8 |
| | Lift Dragon | 10 | 0 | 1 | 2.5 | 7 | 7 |
| | Open Plastic Jar (Twist) | 10 | 1 | 1 | 2 | 5.5 | **6** |
| **Semantic Gen** | Move Skull into Drying Rack | 10 | 1 | 0 | 5 | 5 | 5 |
| | Lift Solid Glue | 10 | 3 | 0 | 0 | 2 | 2 |
| | Take Meat Slice out of Brown Plate | 10 | 6 | 0 | 5 | 5 | 5 |
| | Stack Green Cup on Yellow Cup | 10 | 0.5 | 0 | 5.5 | 5 | **5.5** |
| **Language Grounding** | Put {Red, Green} Block into Basket | 10 | 2.5 | 4 | 8.5 | 7.5 | **8** |
| | Lift {Green Block, Yellow Dinosaur} | 10 | 1.5 | 2.5 | 8.5 | 10 | 10 |
| | Put {Red, Green} Block on Plate | 10 | 5 | 5.5 | 8.5 | 9.5 | **10** |
| | Bring Either Pen or Pencil | 10 | 2 | 3 | 6 | 8 | 8 |
| **Mean Success Rate (%)** | | | 18.5±2.7 | 20.0±2.6 | 50.6±3.5 | 70.6±3.2 | **71.1±3.0** |

