# OpenReview forum: "DFA-VLA: Enhancing Robotic Manipulation via Embodied Intelligence"
_ICLR.cc/2026/Conference — Submitted to ICLR 2026_

### Official Review · Reviewer_515L · 2025-10-28

**Soundness:** 2
**Presentation:** 1
**Contribution:** 2
**Rating:** 2
**Confidence:** 3

**Summary:**

This paper proposes DFA-VLA, an end-to-end Vision-Language-Action model that targets two core weaknesses of current robotic manipulation systems: insufficient fine-grained visual understanding and rigid, static cross-modal fusion. Across LIBERO benchmarks and real-world robot deployments, DFA-VLA achieves higher success rates and lower latency than state-of-the-art baselines.

**Strengths:**

- Identifies a core problem of current VLA models: poor modeling of fine-grained visual elements.
- Provides detailed method description.

**Weaknesses:**

- The method is overly complex yet yields only modest gains (~1%) in both simulation and real-world evaluations.
- The draft is poorly prepared and difficult to read. Figures are misaligned, and font sizes are inconsistent. Many citations are missing or incorrectly formatted (e.g., use \citep for parenthetical and \citet for narrative).
- The real-world evaluation is limited and insufficiently described. Include hardware and environment figures and provide detailed task descriptions, protocols, metrics, and failure cases to make the evaluation reproducible and convincing.

**Questions:**

Please refer to Weaknesses.

---

### Official Review · Reviewer_CfRd · 2025-10-28

**Soundness:** 3
**Presentation:** 1
**Contribution:** 2
**Rating:** 2
**Confidence:** 5

**Summary:**

This paper proposes a method to handle fine-grained visual elements and dynamic cross-modal alignment. To this regard, the Dynamic Finegrained Alignment-based Vision-Language-Action (DFA-VLA) is proposed with two main components: 1) Multi-scale Visual-Semantic Modeling (MVSM) Module to extract high-resolution semantic features to boost perception of small objects, occlusions and cluttered backgrounds, and 2) the Dynamic Fine-grained Alignment and Fusion (DFAF) Module to algn language-visual features with mask-guided sparse dynamic attention and gating network. Evaluation on LIBERO benchmarks and real-wold settings shows the effectiveness of the proposed DFA-VLA.

**Strengths:**

1. The results on LIBERO demonstrate that the proposed method outperforms both Diffusion Policy and OpenVLA, which were considered state-of-the-art last year.

2. Ablation study on MVSM and DFAF module verified their effectiveness to some extent.

**Weaknesses:**

1. The title should be revised to provide more informative content, specifically by explicitly defining what DFA-VLA entails.

2. Lack of novelty. The integration of segmentation models within VLA is not a novel concept. Furthermore, the simultaneous use of DinoV2 and SAM may significantly increase resource consumption and inference time. While the authors claim that the paper focuses on enhancing timeliness, there is insufficient computational time analysis to substantiate this assertion. Although latency results are presented in Tables 2 and 3, these do not adequately reflect the actual time required for processing.

3. In the abstract, the authors assert that MVSM enhances the perception of small objects, occlusions, and cluttered backgrounds. However, there is a lack of experimental results to support these claims.

4. Poor presentation. The manuscript contains citation errors, including missing indices (e.g., line 152 on page 3 and line 265 on page 5). Additionally, several notations, such as H_text and \blod(H) in line 153, and the variable v in Equation 15, are not clearly defined, which could lead to confusion for the reader.

5. Lack of comparison with recent SoTAs. Given the rapid advancements in this field, both diffusion policy and OpenVLA have been surpassed by more recent works. For example, paper [a] shows the avg. SR on LIBERO has been over 80%. Moreover, as indicated in Table 3, the baseline model without the proposed contributions outperforms the Diffusion Policy (82.3% vs. 78.3%), making it difficult to ascertain the true benefits of the proposed method.

[a] He, Yuxin, and Qiang Nie. "ManiTrend: Bridging Future Generation and Action Prediction with 3D Flow for Robotic Manipulation." arXiv preprint arXiv:2502.10028 (2025).

6. The manuscript does not provide a thorough or insightful analysis of the experimental results, which is crucial for understanding the implications of the findings.

**Questions:**

1. Given that DinoV2 and SAM are models that do not incorporate semantic context, how the authors crop task-relevant regions from V_global and how the fine-grained alignment is implemented?

2. What exactly is the action policy employed in this work? Providing detailed information on this aspect is crucial for understanding its implications in robotic manipulation.

---

### Official Review · Reviewer_4yUJ · 2025-10-31

**Soundness:** 1
**Presentation:** 2
**Contribution:** 1
**Rating:** 2
**Confidence:** 4

**Summary:**

This paper aims to address two issues in Vision-Language-Action (VLA) models: limited fine-grained perception and static cross-modal fusion. It proposes DFA-VLA, which integrates a multi-scale visual-semantic modeling module and a dynamic fine-grained alignment and fusion mechanism.
Experiments demonstrate performance improvements on libero benchmark tasks.

**Strengths:**

The paper has a well-structured framework and achieves performance that surpasses OpenVLA.

**Weaknesses:**

1. This paper mentions two motivations:  poor modeling of fine-grained visual elements and over-reliance on static cross-modal attention, restricting adaptability and generalization in complex open environments. However,  there is no direct experiment, visualization, or mathematical formulation to validate these motivations. The comprehensive scores on a certain benchmark are hard to validate these motivations. I highly recommend that authors further delve into their motivations and validate them with more visualizations.

2. For the second motivation, I also agree that how to balance visual and linguistic information is a critical issue. However, the term 'static' is quite confusing. What are the differences between 'static' and 'dynamic'?

3. The two modules of this paper seem to be a simple engineering combination.

4. The comparison with other SOTA methods is insufficient, like Pi 0 or 0.5.

5. The term 'Embodied Intelligence' in the title is also very confusing, since it carries little informational relevance to the actual ideas presented in this paper.

**Questions:**

Please see the weaknesses.

---

### Official Review · Reviewer_6nUu · 2025-10-31

**Soundness:** 1
**Presentation:** 1
**Contribution:** 2
**Rating:** 0
**Confidence:** 4

**Summary:**

This paper introduces DFA-VLA, a Vision-Language-Action (VLA) model designed to improve robotic manipulation by addressing two key limitations of existing models: (1) poor modeling of fine-grained visual details and (2) over-reliance on static cross-modal attention. The proposed architecture is built on a Llama 2 7B backbone  and incorporates two novel components: the Multi-scale Visual-Semantic Modeling (MVSM) module, which combines features from DinoV2 and SAM to create a rich, multi-scale visual representation , and the Dynamic Fine-grained Alignment and Fusion (DFAF) module, which claims to use sparse dynamic attention and a text-driven gating mechanism for more efficient and adaptive vision-language fusion. The authors evaluate their model on the LIBERO simulation benchmark and in real-world sim-to-real experiments, claiming state-of-the-art (SOTA) performance in both success rates and inference efficiency.

**Strengths:**

The paper targets a well-recognized and important problem in the VLA domain. The limitations of static fusion and the need for better fine-grained visual understanding, especially for small objects and occlusions, are significant hurdles for robust robotic manipulation. The motivation to create a more dynamic and perceptive model is commendable. Leveraging SAM and DinoV2 is practical and intuitive for fine-grained perception tasks.

**Weaknesses:**

This paper suffers from critical flaws in the clarity of its core contribution and the significance of its results, which undermine its claims.

1. **Critical Lack of Clarity in the Core Method (DFAF):** The DFAF module is presented as the central methodological innovation, but its description in Section 3 and Figure 2 is profoundly confusing, contradictory, and non-reproducible.
    * **Incomprehensible Equations:** Equations (8)-(10)  are vague and appear dimensionally incorrect. For instance, Equation (9), $V_{sparse} = \text{TopK}(\cos(H_{text}, V_{proj}))$, is ill-defined. $H_{text}$ and $V_{proj}$ are matrices of features; taking the cosine similarity results in a 2D similarity matrix. It is unclear what "TopK" of this matrix means. Does it select K values? K rows? K columns? How is this used to "sparsify the visual tokens"?  Equation (10)  is equally confusing, showing an addition between $V_{sparse}$ (of unknown shape) and a broadcasted text vector, without a clear explanation of how their dimensions are made compatible.
    * **Contradictory Diagram:** Figure 2  contradicts the text. The text states the DFAF module is applied "For the first three blocks" of the Llama decoder. However, Figure 2 depicts $X$ being fed into "Llama 2 7B (First 3 layers)" to produce $H_{text}$, while a *separate* "Llama 2 7B (Backbone)" branch produces $V_{sparse}$. This implies two parallel backbones, which is not what the text describes.
    * **Overall:** This lack of clarity makes the paper's primary contribution impossible to verify, evaluate, or reproduce.

2.  **Marginal (or Non-Existent) Performance Improvement:** The paper's central claim of "outperform[ing] state-of-the-art methods"  is a significant overstatement not supported by the data.
    * **LIBERO Results (Table 1):** The performance gains over OpenVLA are statistically minimal. On LIBERO-Spatial, the gain is 0.8% (85.5% vs 84.7%). More importantly, the paper's claim of being "especially" good at "long-term tasks"  is **directly contradicted** by Table 1, where DFA-VLA (53.7%) performs **significantly worse** than OpenVLA (56.5%) on the LIBERO-Long benchmark.
    * **Real-World Results (Figure 3 & Table 4):** The average real-world success rate is 71.1% for DFA-VLA versus 70.6% for OpenVLA. This 0.5% difference is negligible and well within the reported standard error (3.0% and 3.2% respectively ), meaning the results are statistically indistinguishable. Furthermore, the bar chart  shows DFA-VLA performs *worse* than OpenVLA on "Visual Generalization" and "Motion Generalization".

3.  **Insufficient Ablation Study due to Weak Baseline:** The ablation study (Table 3) is insufficient for properly evaluating the paper's novel contribution (DFAF).
    * The study's baseline (Model 1) uses a generic ViT-Base , which achieves only 82.3% SR. This is a "strawman" baseline, as it is significantly weaker than the SOTA OpenVLA baseline (84.7% SR).
    * The study shows that adding MVSM (Model 2) brings performance to 84.7% , which is *exactly* the same as OpenVLA's performance.
    * This design is insufficient because it fails to isolate the true contribution of the *novel* DFAF module. The baseline for evaluating DFAF should have been a strong, SOTA-equivalent model (i.e., Model 2 or OpenVLA).

4. **Poor Presentation and Writing Quality:** The paper's quality is severely undermined by numerous presentation issues, suggesting a rushed and careless submission.
    * Figures 1 and 2  are low-resolution, poorly labeled, and visually confusing, hindering the understanding of the MVSM and DFAF modules they are meant to illustrate.
    * Some passages read like product documentation or future-work brainstorming, not scientific argument. Paper includes aspirational but vague statements (e.g., "task grammar trees", "unifying task levels") that detract from crisp contribution focus.

**Questions:**

1.  **DFAF Module Clarity:** Can the authors please provide a complete and unambiguous definition of the DFAF module, preferably with pseudo-code?
    * Specifically, what are the exact dimensions and operations in Eq. (9) ? How is $V_{sparse}$ derived from the similarity matrix $\cos(H_{text}, V_{proj})$?
    * How are the terms in Eq. (10)  (i.e., $V_{sparse}$ and the broadcasted mean of $H_{text}$) made dimensionally compatible for addition?
    * Can you resolve the contradiction between the text (DFAF applied *to* first 3 layers)  and Figure 2 (parallel backbones)?

2.  **Performance Claims:** How do you reconcile the claim in the abstract of being strong in "long-term tasks"  with the results in Table 1, which show DFA-VLA underperforming OpenVLA by nearly 3% on LIBERO-Long?

3.  **Statistical Significance:** The 0.5% average improvement in the real-world evaluation (71.1% vs 70.6%) is much smaller than the reported standard error (~3.0%). Do you have any statistical tests to demonstrate that this is a statistically significant improvement and not experimental noise?

4.  **Ablation Baseline:** Why was a weak ViT-Base (Model 1)  chosen as the primary baseline for the ablation study, rather than a stronger, SOTA-equivalent vision backbone? Do you agree that using Model 2 (84.7% SR)  as the baseline for evaluating DFAF would be more appropriate, and that this shows your novel module's contribution is only 0.8%?

Above all, I don't think this paper meets the bar and I suggest reject.

---

### Meta-Review · Area_Chair_SFdJ · 2026-01-10

**Summary:**

Reviewers agree that the paper tackles a relevant problem in VLA, but identify serious shortcomings that prevent acceptance. The core DFAF module is insufficiently specified, with unclear equations, inconsistent descriptions, and contradictions between text and figures, raising reproducibility concerns. Empirical gains over strong baselines are marginal, often within variance, and sometimes worse on long-horizon tasks. Key claims on fine-grained perception, dynamic fusion, efficiency, and long-term reasoning are not directly validated. Ablation studies rely on weak baselines and do not isolate contributions. Presentation quality is consistently poor.

**Reviewer Concerns:**

No rebuttal was provided, and therefore reviewer concerns remain unaddressed.

**Reviewer Scores:**

Reviewer 1 would very likely maintain a strong reject due to unresolved soundness and reproducibility issues. Reviewers 2 and 4 might acknowledge minor clarifications but would still recommend reject, given weak validation, limited novelty, and marginal gains. Reviewer 3, despite a relatively higher soundness score, would likely keep a reject recommendation due to lack of novelty, unclear action policy, insufficient analysis, and missing comparisons with recent SOTA work. Overall, no reviewer is expected to move to a borderline or positive recommendation after discussion.

---

### Decision · Program_Chairs · 2026-01-26

Reject